# Importation, Local Transmission, and Model Selection in Estimating the Transmissibility of COVID-19: The Outbreak in Shaanxi Province of China as a Case Study

**DOI:** 10.3390/tropicalmed7090227

**Published:** 2022-09-03

**Authors:** Xu-Sheng Zhang, Huan Xiong, Zhengji Chen, Wei Liu

**Affiliations:** 1Statistics, Modelling and Economics, Data, Analytics & Surveillance, UK Health Security Agency, London NW9 5EQ, UK; 2School of Public Health, Kunming Medical University, Kunming 650500, China

**Keywords:** basic reproduction number, Bayesian inference, COVID-19, mathematical modelling, model selection, local transmission, importation

## Abstract

**Background**: Since the emergence of the COVID-19 pandemic, many models have been applied to understand its epidemiological characteristics. However, the ways in which outbreak data were used in some models are problematic, for example, importation was mixed up with local transmission. **Methods**: In this study, five models were proposed for the early Shaanxi outbreak in China. We demonstrated how to select a reasonable model and correctly use the outbreak data. Bayesian inference was used to obtain parameter estimates. **Results**: Model comparison showed that the renewal equation model generates the best model fitting and the Susceptible-Exposed-Diseased-Asymptomatic-Recovered (SEDAR) model is the worst; the performance of the SEEDAR model, which divides the exposure into two stages and includes the pre-symptomatic transmission, and SEEDDAAR model, which further divides infectious classes into two equally, lies in between. The Richards growth model is invalidated by its continuously increasing prediction. By separating continuous importation from local transmission, the basic reproduction number of COVID-19 in Shaanxi province ranges from 0.45 to 0.61, well below the unit, implying that timely interventions greatly limited contact between people and effectively contained the spread of COVID-19 in Shaanxi. **Conclusions**: The renewal equation model provides the best modelling; mixing continuous importation with local transmission significantly increases the estimate of transmissibility.

## 1. Introduction

The emerging coronavirus disease, COVID-19, has been circulated worldwide since January 2020 [1,2,3]. To control its spread, it is crucial to accurately estimate its important epidemiological characteristics such as transmissibility and to predict its further potential spread under different control measures. For this, mathematics and statistics have been used to model the transmission dynamic processes [4,5,6,7]. To obtain reliable estimates of the epidemiological characteristics from modelling analyses, correctly distinguishing and using different outbreak data in an appropriate transmission model is essential [8,9,10,11,12,13].

The transmissibility of an infectious agent describes how easy and fast an infectious disease can spread within a population. It is usually measured by the basic reproduction number (denoted as *R*_0_), which is defined as the average number of secondary infections generated by an infectious person introduced into a completely susceptible population [5]. Although many methods of estimating *R*_0_ have been developed [14,15], the difficulty in measuring *R*_0_ of COVID-19 lies in the fact that it is a novel coronavirus. The knowledge of the well-known coronaviruses such as severe acute respiratory syndrome (SARS) and Middle East respiratory syndrome (MERS) has been borrowed to understand the early transmission dynamics of COVID-19 [11]. Nevertheless, the epidemiological characteristics of COVID-19 appear quite different from those of both SARS and MERS [16]. Further, as R_0_ is determined by the infectiousness of SARS-CoV-2 and the contact rate between individuals, its value should be different among regions that implemented different control measures. Therefore, the basic knowledge of COVID-19 epidemiological features should be obtained from the epidemic data during outbreaks.

During the initial outbreaks of COVID-19 in China from January to March 2020, the national and provincial governments and public health authorities collected lots of data about the outbreaks and individual cases. These data undoubtedly provide a good chance for us to understand the transmissibility of COVID-19 and the impact of control measures implemented on stopping the spread. The reliable and accurate estimates depend on our understanding of how SARS-CoV-2 is transmitted in the population and the appropriate inference methods to calibrate the transmission models. We noticed that although some modelling studies have been published [17,18,19,20,21,22,23,24], they might have problems in obtaining reliable estimates of epidemiological parameters because of inappropriate use of the outbreak data. For example, when estimating *R*_0_, those studies used the daily number of cases, which implicitly summarised local and imported cases. During the early stages of the COVID-19 pandemic, one common feature among the outbreaks, except that in the epicentre, Wuhan city, was the continuous importation due to quick and easy modern transportation. During the outbreaks, the role played by imported cases is different from that of local cases when counting the transmissibility of SARS-CoV-2: local cases as a result of local transmission can increase *R*_0_, while imported cases as a potential source of transmission should reduce the estimate of *R*_0_. A previous modelling study of the spatial transmission of pandemic flu [25] shows that early importation plays a relatively more important role in estimating transmissibility. Models that mixed up continuous importation with local transmission enlarged the estimate of *R*_0_ [18,19,20,21,22,23,24,26], and mislead our assessment. To get a reliable estimate of *R*_0_, it is crucial to separate imported cases from local cases [13,21,26,27,28,29].

Many different models have been proposed to describe the transmission dynamics of COVID-19, such as compartmental transmission dynamics models [8,9,10,11,12,13,28,30,31], the renewal equation model [32], machine learning [31], the Richards growth model [33,34], and time series models, such as the ARIMA model [35]. In theory, we need to ask: which one is better to approximate the spread of infection among the population given the data collected? In practice, we must search for the one that can provide a simple and accurate tool for us to estimate the essential epidemiological parameters and predict the trend of spread within the population.

In this study, we took the COVID-19 outbreak from January to February 2020 in Shaanxi province, China as an example to show how to avoid the common pitfall in estimating *R*_0_ during the early stage of the COVID-19 pandemic in mainland China and other similar situations and show how to select the best transmission model by comparing their fitting to outbreak data. Two models [17,24] have been used for analysing the Shaanxi outbreak. Bai et al. [17] proposed a Susceptible-Exposed-Diseased-Asymptomatic-Recovered (SEDAR) compartmental transmission model, and Yang et al. [24] used the Richards growth model. The same implicit and problematic assumption in their modelling is that the Shaanxi outbreak was caused by one importation event at the very beginning of the outbreak. Based on this assumption, they obtained nearly the same estimate of the basic reproduction number (2.95 and 3.11, respectively). In view of these two models, we will propose five models to analyze the Shaanxi outbreak: the Richards growth model, the renewal equation model, the SEDAR model, the SEEDAR model in which the exposure interval in SEDAR is divided into two with the latter one being infectious, and the SEEDDAAR model in which not only two exposed classes are present as in the SEEDAR model but there are two classes in both diseased and asymptomatic infections, so infectious periods following gamma-distributions. As we show below, the estimate of the basic reproduction number of COVID-19 during the Shaanxi outbreak under the actual continuous importation is below the critical level, 1.0, which is the consequence of the timely and draconian control measures implemented in Shaanxi province.

Although the Shaanxi outbreak was relatively small over a short period of about one month and might be out of date, it was a typical situation of local outbreaks in mainland China except for the epicentre, Wuhan city, and it represented many similar situations in other countries during the early phases of COVID-19 pandemic. It is therefore hoped that this study provides some useful modelling methods in dealing with similar outbreak situations in future.

## 2. Materials and Methods

### 2.1. Data

The outbreak data for COVID-19 were collected from the Shaanxi provincial government website from 23 January to 20 February 2020. Variables used in the line list data for COVID-19 included age, gender, place of origin, exposure date, symptom onset date, hospital admission date, close contacts, cluster number, medical history, symptoms, and travel history. The serial interval and incubation period are estimated by applying the R language function *fitdistr* to the line list data of the Shaanxi outbreak. Among 245 cases reported during the period, 113 were imported from outside of Shaanxi (Figure 1A). The dates of symptom onset were recorded for 210 cases, from which the delay from symptom onset to reporting was estimated to have a mean of 7.54 days and a standard deviation of 4.12 days. The other 35 cases whose dates of symptom onset were missed were imputed from their reporting dates and the distribution of delays from symptom onset to reporting. The timeline of dates of symptom onset of 245 cases is shown in Figure 1B.

To provide the direct dates for local transmission modelling, we constructed modified dates of symptom onset for imported cases. If the date of symptom onset of one imported case was earlier than the date of entry into Shaanxi province, then its modified date of symptom onset is its entry date; otherwise, the modified date of symptom onset is its date of symptom onset. There are 19 cases whose dates of the symptom onset were earlier than their entry dates. This modification of symptom onset dates of imported cases will help in modelling the transmission within Shaanxi province. With these arrangements, the timeline of modified dates of symptom onset is shown in Figure 1C.

Once person-to-person transmission of COVID-19 was confirmed and announced on 20 January 2020 in mainland China, the Shaanxi provincial government took rapid actions to launch control measures for COVID-19 containment from 21 January 2020 [36]. The measures implemented in Shaanxi included strict traffic health quarantine, strictly limiting public gathering activities, timely and effective medical treatments, overall coordination of personnel and material allocation, timely release of information according to the law, strengthening publicity and education, professional training, and resolutely safeguarding social stability. These measures effectively controlled the local transmission and quickly reduced the number of importations, as reflected in the epidemic curve shown in Figure 1.

### 2.2. Models

We proposed five models for analysing the SARS-CoV-2 outbreak in Shaanxi: the Richards growth model, the renewal equation model, and three compartmental models: the Susceptible-Exposed-Diseased-Asymptomatic-Recovered (SEDAR), Susceptible-Exposed-Exposed-Diseased-Asymptomatic-Recovered (SEEDAR), and Susceptible-Exposed-Exposed-Diseased-Diseased-Asymptomatic-Asymptomatic-Recovered (SEEDDAAR) models. Bayesian inference via Markov chain Monte-Carlo (MCMC) sampling was used to estimate *R*_0_ by calibrating the five models to Shaanxi outbreak data. The details of models and inference methods are given below.

#### 2.2.1. Richards Growth Model

The Richards growth model is an extended form of logistic growth model, an ecological population growth model used to describe the growth of a population under competition for resources due to carrying capacity [37,38]. It has been widely used in population biology, including infectious disease dynamics [32,33]. For an outbreak caused by *C*_0_ seeds of infection at time *t*_0_, the Richards growth model states that the cumulative number of cases at time *t* is given by the following equation [24,33,39]:$$C\left(t\right)=K{\left(1+\left({\left(\frac{K}{C_{0}}\right)}^{\nu }-1\right)\exp\left(-r\nu \left(t-t_{0}\right)\right)\right)}^{-\frac{1}{\nu }}.$$

Here, *r* is the growth rate, *ν* is the scaling exponent, and *K* is the final epidemic size given *C*_0_ = *C*(*t*_0_) seeds. If importation is continuing (e.g., there are *C_i_* cases that are imported at *t_i_*, *i* = 0,…, *n* − 1) and the outbreaks that importation at different times can cause are of the same final size *K* and growth rate *r*, then the total cumulative number of cases should be summarized as:(1)$$C\left(t\right)=K\overset{n-1}{\underset{i=0}{\sum }}H\left(t-t_{i}\right){\left(1+\left({\left(\frac{K}{C_{i}}\right)}^{\nu }-1\right)\exp\left(-r\nu \left(t-t_{i}\right)\right)\right)}^{-\frac{1}{\nu }}.$$

Here, H(*t* − *t_i_*) is the Heaviside function: which is 1 if *t* > *t_i_* and 0 otherwise. The daily number of new local cases can be calculated as *μ*(*t*) = *C*(*t*) − *C*(*t* − 1). The basic reproductive number *R*_0_ can be calculated from the growth rate and serial interval which is assumed to follow gamma-distribution g(τ;α,β) by [40,41]
(2)$$R_{0}=\frac{1}{\int_{0}^{\infty }g\left(\tau ;\alpha ,\beta \right)e^{-r\tau }dx}={\left(1+\frac{r}{\beta }\right)}^{\alpha }.$$

#### 2.2.2. Renewal Equation Model

It is assumed that, once infected, individuals have an infectivity profile given by a probability distribution *w*_s_, dependent on time since infection of the case, *s*, but independent of calendar time, *t*. The distribution *w*_s_ typically depends on individual biological factors such as pathogen shedding or symptom severity. For simplicity, the distribution *w*_s_ is approximated by the distribution of serial interval (SI), the lag in onset dates of symptoms between an infector and its infectee. In the original renewal equation model, Fraser [42] considers a situation where the only importation is index case(s) at the very beginning of the outbreak and other cases are generated by local transmission (this assumption was also made in its direct application software for estimating the time-varying reproduction number [43]). During the spread of COVID-19 in 2020, the outbreak within a region (except the epicentre, Wuhan) took place with continuous importation. To take this into account, Fraser’s model is slightly modified as in the following (c.f., [44]). Let *c_t_* be the number of local cases whose symptoms onset at day *t*, its expected value is approximated by:(3)$$E\left(c_{t}\right)=R_{0}\sum_{j=1}^{\min\left(t-1,SI\_max\right)}w_{s}(c_{t-s}+I_{t-s}).$$

Here, *I_t_*_−*s*_ is the number of imported cases that have the onset date of symptoms on day *t*_−_*s* and *w_s_* represents the probability mass function of the SI of length *s* days, which can be obtained by $w_{s}=G\left(s\right)-G\left(s-1\right)$, with *G*(.) representing the cumulative distribution function of the gamma distribution. The gamma distribution is characterized by its mean SI_mean and standard deviation SI_sd, both of which are to be estimated jointly with *R*_0_ from the outbreak data [45]. Because only 19 cases among 113 imported cases had symptom onset before entering Shaanxi province, the assumption that all cases started their infectivity duration within Shaanxi province, which is implicitly required in Equation (3), should be approximately satisfied.

In Equation (3) an implicit assumption made is that the transmissibility (i.e., *R*_0_) remained constant during the outbreak duration. This should be reasonable in view of the timely control measures implemented in Shaanxi province: control measures started on 21 January 2021 [36] and raised to their first-class emergency responses on 25 January 2020, just 2 days after the reporting of the first three imported cases in Shaanxi province [16,23]. To estimate the daily-varying transmissibility *R_t_*, Equation (3) is rearranged as:$$R_{t}=c_{t}/\sum_{j=1}^{\min\left(t-1,SI\_max\right)}w_{s}(c_{t-s}+I_{t-s}).$$

That is, *R_t_* can be estimated by the ratio of the number of new infections produced at time step *t*, *c*_t_, to the total infectiousness of infected individuals at time *t*, given by $\sum_{j=1}^{\min\left(t-1,SI\_max\right)}w_{s}(c_{t-s}+I_{t-s})$, the sum of infection incidence, including both imported and locally generated, up to time step *t*−1 or the maximum of SI (whichever is the smallest), weighted by the infectivity function *w*_s_. *R*_t_ is the average number of secondary cases that each infected individual would infect if the conditions remained as they were at time *t* [43], and it is used to monitor the change in transmissibility along the course of an outbreak.

#### 2.2.3. SEDAR Transmission Model

Figure 2A shows the schematic for the SEDAR compartmental model: susceptible individuals (*S*) contract SARS-CoV-2 virus from infectious people and then it enters the latent class (*E*); a fraction (*θ*) of those exposed after an average latent period (*L*_1_) progress to become diseased (*I*) and the other fraction (1 − *θ*) remains asymptomatic (*A*) but becomes infectious after an average latent period (*L*_2_). The diseased infections will be detected and admitted to hospital and isolated from the community after an average period of *D*_1_ and the asymptomatic cases recover after an average infectious period of *D*_2_. The model can be described by the following set of differential equations:$$\frac{d}{dt}S\left(t\right)=-\beta S\left(t\right)\left(I\left(t\right)+\xi A\left(t\right)\right)/N$$
$$\frac{d}{dt}E\left(t\right)=\beta S\left(t\right)\left(I\left(t\right)+\xi A\left(t\right)\right)/N-\theta E\left(t\right)/L_{1}-\left(1-\theta \right)E\left(t\right)/L_{2}$$
(4)$$\frac{d}{dt}I\left(t\right)=\theta E\left(t\right)/L_{1}-I\left(t\right)/D_{1}+Imported\left(t\right)$$
$$\frac{d}{dt}A\left(t\right)=\left(1-\theta \right)E\left(t\right)/L_{2}-A\left(t\right)/D_{2}$$
$$\frac{d}{dt}R\left(t\right)=I\left(t\right)/D_{1}+A\left(t\right)/D_{2}$$

Here, *N* is the size of the population under investigation (*N* = 37,330,000 for Shaanxi province) and is assumed to be constant during the outbreak. The definitions of model parameters are given in Table 1. Importantly, the model includes an item for imported cases (i.e., *Imported*(*t*) in equation for *I*(*t*)) from outside of the population as reported [16]. This is to treat imported cases as the source rather than the results of local transmission from the region under investigation, therefore removing the importation as a result of the local transmissibility.

The steady-state solution of the equation system (4) can be easily obtained. The expression for *S** (the size of the population susceptible to infection at equilibrium) is:$$S^{*}=N\frac{\theta /L_{1}+\left(1-\theta \right)/L_{2}}{\beta [\theta D_{1}/L_{1}+\xi \left(1-\theta \right)D_{2}/L_{2}]}.$$

From this, we can obtain the expression of basic reproduction number:(5)$$R_{0}N/S^{*}=\beta [\theta D_{1}/L_{1}+\xi \left(1-\theta \right)D_{2}/L_{2}]/[\theta /L_{1}+\left(1-\theta \right)/L_{2}]$$

Ref [14,30]. In the special situation where *L*_1_ = *L*_2_, expression (5) reduces to:$$R_{0}=\beta \left[\theta D_{1}+\xi \left(1-\theta \right)D_{2}\right].\;$$

#### 2.2.4. SEEDAR Transmission Model

Ferretti et al. [46] show that 30% to 50% of all transmissions are pre-symptomatic transmissions. To take the pre-symptomatic transmission into account, we modify the above SEDAR model by including a secondary exposure compartment (see Figure 2B). For simplicity, this new compartment is assumed to be asymptomatic but of the same infectivity as the symptomatic infections. The corresponding equations are modified as:$$\frac{d}{dt}S\left(t\right)=-\beta S\left(t\right)\left(E_{2}\left(t\right)+I\left(t\right)+\xi A\left(t\right)\right)/N,$$
$$\frac{d}{dt}E_{1}\left(t\right)=\beta S\left(t\right)\left(E_{2}\left(t\right)+I\left(t\right)+\xi A\left(t\right)\right)/N-\theta E_{1}\left(t\right)/L_{1}-\left(1-\theta \right)E_{1}\left(t\right)/L_{2},$$
$$\frac{d}{dt}E_{2}\left(t\right)=\theta E_{1}\left(t\right)/L_{1}-E_{2}\left(t\right)/L_{3},$$
(6)$$\frac{d}{dt}I\left(t\right)=E_{2}\left(t\right)/L_{3}-I\left(t\right)/D_{1}+Imported\left(t\right),$$
$$\frac{d}{dt}A\left(t\right)=\left(1-\theta \right)E_{1}\left(t\right)/L_{2}-A\left(t\right)/D_{2},$$
$$\frac{d}{dt}R\left(t\right)=I\left(t\right)/D_{1}+A\left(t\right)/D_{2}.$$

Compared with the SEDAR model, a new parameter *L*_3_, the duration of the late incubation period in which the infected person can pass the virus on, is introduced and is to be estimated (See Table 1).

Similarly, the basic reproduction number *R*_0_ for the SEEDAR model can be obtained by deriving the expression of the equilibrium number of susceptible people, and it is given by:(7)$$R_{0}=\beta \left[\theta (L_{3}+D_{1}\right)/L_{1}+\xi \left(1-\theta \right)D_{2}/L_{2}]/[\theta /L_{1}+\left(1-\theta \right)/L_{2}].$$

#### 2.2.5. SEEDDAAR Transmission Model

In view of the empirical observations that the infectious period follows the gamma distribution rather than the usual exponential distribution [47,48,49,50,51,52], we introduce the intermediate compartments by evenly dividing diseased compartment *I*(t) into *I*_1_(t) and *I*_2_(t), and dividing asymptomatic compartment *A*(t) into *A*_1_(t) and *A*_2_(t). Adding the two new compartments to Equation (6), the model equations for the SEEDDAAR model are given as:$$\frac{d}{dt}S\left(t\right)=-\beta S\left(t\right)\left(E_{2}\left(t\right)+I_{1}\left(t\right)+I_{2}\left(t\right)+\xi (A_{1}\left(t\right)+A_{2}\left(t\right))\right)/N,$$
$$\begin{array}{l} \frac{d}{dt}E_{1}\left(t\right)\\ =\beta S\left(t\right)\left(E_{2}\left(t\right)+I_{1}\left(t\right)+I_{2}\left(t\right)+\xi (A_{1}\left(t\right)+A_{2}\left(t\right))\right)/\\ N-\theta E_{1}\left(t\right)/L_{1}-\left(1-\theta \right)E_{1}\left(t\right)/L_{2} \end{array},$$
$$\frac{d}{dt}E_{2}\left(t\right)=\theta E_{1}\left(t\right)/L_{1}-E_{2}\left(t\right)/L_{3},$$
(8)$$\frac{d}{dt}I_{1}\left(t\right)=E_{2}\left(t\right)/L_{3}-2I_{1}\left(t\right)/D_{1}+Imported\left(t\right),$$
$$\frac{d}{dt}I_{2}\left(t\right)=2I_{1}\left(t\right)/D_{1}-2I_{2}\left(t\right)/D_{1},$$
$$\frac{d}{dt}A_{1}\left(t\right)=\left(1-\theta \right)E_{1}\left(t\right)/L_{2}-2A_{1}\left(t\right)/D_{2},$$
$$\frac{d}{dt}A_{2}\left(t\right)=2A_{1}\left(t\right)/D_{2}-2A_{2}\left(t\right)/D_{2},$$
$$\frac{d}{dt}R\left(t\right)=2I_{2}\left(t\right)/D_{1}+2A_{2}\left(t\right)/D_{2}.$$

The inclusion of additional compartments in diseased and asymptomatic infections does not change the expression of the basic reproduction number, and the SEEDDAAR model has its basic reproduction number as in Equation (7).

### 2.3. Inference Method by Calibration to Shaanxi Outbreak

Inference is carried out within the Bayesian framework [53,54], obtained through the combination of the prior distributions and the likelihood function. We denote the set of model parameters to be inferred as Θ = {*r*, *ν*, *K*} for the Richards growth model, Θ = {*R*_0_, SI_mean, SI_sd} for the renewal equation model, Θ = {*β*, *L*_1_, *D*_1_} for the SEDAR model and Θ = {*β*, *L*_1_, *L*_3_, *D*_1_} for the SEEDAR and SEEDDAAR models under the special situation where both asymptomatic and symptomatic infections are of the same latent period and infectious period (i.e., *L*_2_ = *L*_1_ and *D*_2_ = *D*_1_). For simplicity, the proportion of symptomatic infections (*θ*) is set at 98.9% as reported [16]. Given the values of parameters Θ for the Richards growth model and the renewal equation model, simulating the time series of local infections, denoted as *μ*(*t*), *t* = *t*_start_, …, *t*_end_, is straightforward. Here, *t*_start_ and *t*_end_ represent the start day and end day of the outbreak data collected, respectively. For each set of parameter values of SEDAR, SEEDAR, and SEEDDAAR models, the Runge–Kutta fourth order method is used to solve the model equations and to obtain predicted time series of infections. In the inference of model parameters, directly observed cases of modified symptom onset dates (see the definition in the data above) are used as illustrated in the following. The likelihood function for the observed time series of local cases *x*(*t*), *t* = *t*_start_, …, *t*_end_, is given as:$$L(\Theta |\mathrm{Data})=\prod_{t=t_{\mathrm{start}}}^{t_{\mathrm{end}}}\frac{\Gamma \left(x\left(t\right)+r\left(t\right)\right)}{\Gamma \left(r\left(t\right)\right)\Gamma \left(x\left(t\right)+1\right)}{\left(\frac{1}{\eta }\right)}^{r\left(t\right)}{\left(1-\frac{1}{\eta }\right)}^{x\left(t\right)}.$$

Here, $r\left(t\right)=\frac{\mu \left(t\right)}{\eta -1}$ with *η* being the dispersion parameter of the negative binomial distribution. The parameters are estimated using MCMC methods with Gibbs sampling and non-informative flat priors. The boundaries of uniformly distributed priors are set forth as in the literature [16] and the data collected from Shaanxi province (Figure 3). The details of the MCMC sampling method are given below.

#### MCMC Sampling

To propose new values for parameters, we use normal random walk. Suppose the current value of the *j*th parameter of Θ is Θ*_j_*^(t−1)^, the new proposal is:Θ*_j_** = Θ*_j_*^(t−1)^ + σ*_j_*z.

Here, *z* is a standard normal variable and σ*_j_* is the step size of the *j*^th^ parameter. The normal proposal density is given by:$$q(\Theta^{*}|\Theta^{\left(t-1\right)})=\frac{1}{\sigma \sqrt{2\pi }}\exp\left[-\frac{{\left(\Theta^{*}-\Theta^{\left(t-1\right)}\right)}^{2}}{2\sigma^{2}}\right].$$

That is, Θ*_j_** follows *N*(Θ*_j_*
^(t−1)^, σ*_j_*^2^) (normal distribution with mean = Θ*_j_*
^(t−1)^, and standard deviation = σ*_j_*). The proposal is accepted as the next step of the Markov chain with probability α = min(A,1), where:$$A=\frac{\pi \left(\Theta^{*}\right)}{\pi \left(\Theta^{t-1}\right)}\frac{L(\Theta^{*}|y)}{L(\Theta^{\left(t-1\right)}|y)}\frac{q(\Theta^{\left(t-1\right)}|\Theta^{*})}{q(\Theta^{*}|\Theta^{\left(t-1\right)})}.$$

Here, π(.) denotes the prior density, *L*(Θ*_j_**|y) the likelihood of parameter Θ*_j_** given data y. For a truncated normal walk on the range (*a*,*b*), the proposal density is given by:$$q\left(\Theta^{*}{|\Theta }^{\left(t-1\right)}\right){|}_{\left(a,b\right)}=\frac{q\left(\Theta^{*}{|\Theta }^{\left(t-1\right)}\right)}{\Phi \left(\frac{b-\Theta^{\left(t-1\right)}}{\sigma }\right)-\Phi \left(\frac{a-\Theta^{\left(t-1\right)}}{\sigma }\right)}.$$
where Φ(.) is the cumulative distribution function of standard normal. The expression for A is consequently modified as:$$A=\frac{\pi \left(\Theta^{*}\right)}{\pi \left(\Theta^{t-1}\right)}\frac{L(\Theta^{*}|y)}{L(\Theta^{\left(t-1\right)}|y)}\frac{\Phi \left(\frac{b-\Theta^{\left(t-1\right)}}{\sigma }\right)-\Phi \left(\frac{a-\Theta^{\left(t-1\right)}}{\sigma }\right)}{\Phi \left(\frac{b-\Theta^{*}}{\sigma }\right)-\Phi \left(\frac{a-\Theta^{*}}{\sigma }\right)}.$$

Sample a uniformly distributed random number (*r*) between 0 and 1,
Θ*_j_*
^(t)^ = Θ*_j_** if *r* < α (accepted);Θ*_j_*
^(t)^ = Θ*_j_*
^(t−1)^ otherwise (rejected).

To generate nearly independent samples of model parameters, the samples are to be thinned every 400th observation. To respond to the acceptance rate, the following adaptive procedure is applied: if the acceptance ratio over 400 × 200 iterations is less than 12%, then decrease the jump step to 80% of its current size (i.e., σ*_j_* = 0.8σ*_j_*); if it exceeds 40%, then σ*_j_* = 1.2σ*_j_*. Otherwise, the jump step σ*_j_* remains unchanged. To allow the MCMC process to fully converge, a burn-in period of 400,000 iterations is chosen, and the estimates of model parameters are obtained from the further 400,000 iterations.

To compare the performance of the five models [55], the deviance information criterion (DIC), which combines the goodness of fit and model complexity [56], is used. It measures fit via the deviance Dev(Θ) = −2log*L*(Θ|Data) and complexity by an estimate of the ‘effective number of parameters’ *p*_D_ = mean(Dev(Θ)) − Dev(mean(Θ)) (i.e., posterior mean deviance minus deviance evaluated at the posterior mean of the parameters). The DIC is calculated as:DIC = Dev(mean(Θ)) + 2*p*_D_ = mean(Dev(Θ)) + *p*_D_.

The model that has the smallest DIC is the best.

## 3. Results

### 3.1. Estimates of SI and Incubation Period from Line List Data

The results are shown in Figure 3. Fitting the nonnegative data to gamma distributions, the estimates are: From 85 pairs of infector–infectees observed during the outbreak, the SI is estimated to have a mean of 6.29 days and a SD of 4.11 days (so the fitted gamma distribution has a shape parameter of *α* = 2.34 and a rate parameter of *β* = 0.37). From 100 cases that had dates of exposure and symptom onset, the incubation period is estimated to have a mean of 6.76 days and a SD of 4.41 days, and the delay from the onset of symptoms to hospitalization from 222 cases has a mean of 3.71 days and a SD of 2.83 days. If fitting all collected data (including both negative and positive) to normal distributions, the estimates will be shorter (Figure 3). These estimates are consistent with the system review of both the serial interval and the incubation period [57].

### 3.2. Estimate of R_0_ in Shaanxi Outbreak

#### 3.2.1. Richards Growth Model

Model fitting to the daily number of local cases suggests the growth rate *r* = 0.020 with a 95% confidence interval (95% CI): 0.012, 0.032, and the final epidemic size *K* = 3315 (95% CI: 56, 6521) (Table 1). Based on the estimate of gamma-distributed serial interval (Figure 3), the basic reproduction number is calculated using Formula (2) to be 1.13 (95% CI: 1.08, 1.21). We note that although the prediction of the Richards growth model can be fitted to the daily number of local cases within the outbreak period of 30 days, the continuing increase in the daily number of local cases that the model predicts obviously deviated from the actual observations after the outbreak (Figure 4A). That is, even though the model calibration performs well, the external validation is bad. Our estimate of the growth rate is very near to zero, the nonnegative limit. This is in sharp contrast with Yang et al. 2021 [24] who obtained an estimate of growth rate *r* = 0.23 per day by assuming one importation event and mixing up imported cases thereafter with local cases in their modelling study. This points out the limitation of applying the Richards growth model to infection spread processes. In general, infections can either increase, decrease, or remain the same within a population (i.e., *r* can be positive, negative, or zero, with *R*_0_ being larger, less than, or equal to 1.0). Applying the Richards growth model to an infection spread process, it is implicitly assumed that its growth rate is positive and *R*_0_ >1, which is wrong for the situation of well-controlled infections such as COVID-19 in Shaanxi province during January and February of 2020.

#### 3.2.2. Renewal Equation Model

Bayesian inference suggests the basic reproduction number (*R*_0_) has a median of 0.61 and 95% CI from 0.54 to 0.68 (Table 1). The SI is estimated to have a mean of 4.66 days and an SD of 11.73 days, which is shorter than but comparable with the direct observation of SI from the outbreak (Figure 3). The model projection into next month (Figure 4B) indicates that the outbreak will die out within two weeks (i.e., the end of February 2020) and is unlikely to generate any further local cases under the current restriction measures.

The time-varying reproduction number *R*_t_ shown in Figure 5 demonstrates how the transmissibility changed along the course of the outbreak. *R*_t_ increased to about 2.0 within the first week, and then reduced to low values, but occasionally exceeding the critical value of 1.0. Its overall average is 0.61, which is equal to the median of posterior *R*_0_ in the above model fitting. The change of *R*_t_ reflects the stochasticity of transmission events within the Shaanxi outbreak.

#### 3.2.3. SEDAR Model

The calibration of the SEDAR model under the situation with equal incubation and infectious periods for both symptomatic and asymptomatic infection shows that: *R*_0_ is estimated at 0.59 with 95% CI from 0.51 to 0.71 (Table 1). The incubation period is 1.8 days (95% CI: 1.6, 2.9), and the infectious period is 3.8 days (95% CI: 3.5, 5.3). The SEDAR model fitting and its prediction over one month ahead are shown in Figure 4C. Sensitivity analyses (data not shown) show that nearly the same estimates of model parameters are obtained when considering different values of latent period (*L*_2_) and infectious period (*D*_2_) for asymptomatic infections and the relative infectivity (*ξ*) of asymptomatic infections to symptomatic infections. This reflects the fact that, in the Shaanxi outbreak, the asymptomatic infections occupied a very small proportion of all infections (1.1%) [16] and therefore had small effects on model performance.

#### 3.2.4. SEEDAR Model

The *R*_0_ is estimated at 0.45 with 95% CI from 0.30 to 0.76 (Table 1). The incubation period of symptomatic infection is 5.0 days (95% CI: 1.3, 9.7), which is consistent with the observed values (mean = 6.76 days and sd = 4.41 days), and the infectious period of symptomatic infections is 4.8 days (95% CI: 1.6, 14.1), which is longer than the delay from the onset date of symptoms to hospitalization: mean = 3.71 days and sd=2.83 days (Figure 3). The duration of pre-symptomatic transmission (*L*_3_) is estimated at 1.5 days (95% CI: 1.0, 4.4 days); this suggests the fraction of transmission from strictly pre-symptomatic infections was about 1.5/(1.5 + 4.8) =24%, which is in agreement with previous estimates [46]. The SEEDAR model fits well with the observed data and predicts that the outbreak will die out within about three weeks (Figure 4D).

#### 3.2.5. SEEDDAAR Model

The *R*_0_ is estimated at 0.53 with 95% CI from 0.35 to 0.85 (Table 1). The incubation period of symptomatic infection is 5.3 days (95% CI: 1.3, 9.8), and the infectious period of symptomatic infection is 5.4 days (95% CI: 1.7, 14.0). The duration of pre-symptomatic transmission (*L*_3_) is estimated at 1.5 days (95% CI: 1.0 to 4.4 days). Those estimates of model parameters are very similar to those of the SEEDAR model. Similar to the SEEDAR model, the SEEDDAAR model equally well fits the observed data and predicts that the outbreak will die out within about three weeks (Figure 4E).

## 4. Discussion

In this study, five transmission models were proposed to model the COVID-19 epidemic within Shaanxi province of China from early January to late February 2020. By distinguishing imported and local cases in their contribution to local transmission dynamics, we show that the basic reproduction number *R*_0_ of COVID-19 in the Shaanxi outbreak was well below the critical value of 1.0. This indicates that SARS-CoV-2 cannot self-sustain under the current control measures within Shaanxi province, China, and would stop once the importation of COVID-19 cases was halted. Our model successfully predicted the actual epidemic situation in Shaanxi province from late February 2020.

The estimates of *R*_0_ from the renewal equation and the SEDAR models are close to each other, and its estimates from the SEEDAR and SEEDDAAR models are lower; nevertheless, their 95% CIs are closely overlapped. Overall, the estimate of *R*_0_ is in the range from 0.45 to 0.61. The model fittings to the local cases, shown in Figure 4, indicate that the renewal equation model provides the best fit to the observations and the SEDAR model is the worst. This is further confirmed by the values of DIC in Table 1 [55]: 127.9, 175.1, 160.5, and 160.8 for the renewal equation, and the SEDAR, SEEDAR, and SEEDDAAR models, respectively. Furthermore, the better performance of the SEEDAR and SEEDDAAR models than the SEDAR model confirms the existence of pre-symptomatic transmission [46]. It is worth mentioning that the SEEDDAAR model that has its infectious periods following gamma distribution does not appear better than SEEDAR that has its infectious period following simple exponential distribution. The Richards growth model, which is borrowed from ecological population dynamics [37,38], can provide a better model fit to the daily number of local cases than three compartmental transmission models (i.e., SEDAR, SEEDAR, and SEEDDAAR). However, the increasing trend of infections after the first month, which the Richards growth model predicts, deviates from the actual observation and hence invalidates the Richards growth model as an appropriate model for the Shaanxi outbreak.

It is worth emphasizing that although the renewal equation model is the simplest in its structure, it gives the best model fit [52]. Given the distribution of SI of COVID-19, it is straightforward to obtain the estimate of the *R*_0_ [58]. In this study, we perform a joint estimation of *R*_0_ and SI, and the results agree well with three compartmental models in estimation of *R*_0_ and the empirical knowledge of SI [16,57]. Nevertheless, it should be kept in mind that the successful performance of the joint estimation of *R*_0_ and SI in this study may be conditional on the very low proportion (i.e., 1.1%) of asymptomatic infections [58].

One important issue worth pointing out is the continuous importation along the course of an outbreak within one region, except for the epicenter, Wuhan city, China, during the early stages of the COVID-19 pandemic in mainland China. This spectacular feature, unlike previous infectious disease pandemics (such as the 2003 SARS pandemic and the 2009 influenza pandemic), may reflect the rapid and huge movements of modern human beings. If assuming the earliest importation as the only index case(s), the transmission dynamics cannot be appropriately investigated and might be misled [17,24]. Both Bai et al. [17] and Yang et al. [24] also modelled the Shaanxi outbreak but mixed up imported cases with local cases; they obtained estimates of *R*_0_ of about 3.0. Simple reasoning will show that this estimate is problematic. Let us consider a situation where all the 132 cases were generated within Shaanxi province only by the 113 imported cases, a rough estimate is *R*_0_ = 132/113 ≈ 1.2. Some local cases might have been infected by other early local cases rather than directly from imported cases, which implies that the actual *R*_0_ should be less than 1.2. With the similar treatments of continuous importation, the high and problematic estimates of *R*_0_ for outbreaks in the major cities of China (except the epicentre, Wuhan city) were also reported [20,21]. Based on estimates of their SEDAR model parameters, Bai et al. [17] predicted that the Shaanxi outbreak would last until April 2020, which is more than one month longer than the actual occurrence. Our analyses show that the occurrence of the Shaanxi outbreak was mainly due to the large and continuous importation rather than the high local transmissibility of COVID-19 within the province. Furthermore, our prediction is consistent with what happened in Shaanxi province.

In modelling the COVID-19 transmission over the whole of mainland China, we found that *R*_0_ was estimated at 2.23 before 8th February 2020 and then it dropped to 0.04 [30]. Hao et al. [8] also confirmed the effectiveness of the timely prevention and control measures implemented in China in bringing the *R*_0_ well below the critical level of 1.0. To check whether there was any potential breaking point in the transmissibility of COVID-19 within the Shaanxi outbreak, we calculated the instantaneous reproduction number *R*_t_ [43]. The result shown in Figure 5 indicates that no clear pattern emerged that supported a potential breaking point in transmissibility although *R*_t_ exceeded 1.0 on five days. In contrast, having mixed up imported cases with local cases, Yang et al. [24] used the renewal equation method [43] to obtain a time-varying reproduction number (*R*_t_), which persistently decreased over time and stayed beyond the critical level of 1.0 over more than half the course of the outbreak.

Our estimate of *R*_0_ for the Shaanxi outbreak sharply differs from other studies which suggest *R*_0_ = 2–7 [16,49,59,60] for SARS-CoV-2. In theory, *R*_0_ is determined by the infectiousness of SARS-CoV-2 as well as the contact rate between people [5]. In the situation where no vaccine and effective drugs were available to protect people against the virus, the result of *R*_0_ < 1 is due to the highly reduced contact rate between people [30]. This resulted from the timely and strong control measures implemented within Shaanxi province soon after it was announced in public on 20 January 2020 that COVID-19 could be transmitted among people. On the other hand, this indicates the success of the interventions executed in Shanxi province, China.

## 5. Conclusions

Modern inference methodology and mathematical theory can help reveal the unobserved transmission dynamic process and hence provide valuable information for us to understand and control the spread of COVID-19. However, it is important to separate continuous importation from local transmission when modelling the local transmission dynamics of COVID-19. The renewal equation model, albeit being simple in model structure, provides better model fitting and therefore is a practical candidate for analyzing transmission dynamics and monitoring the change in transmissibility.

## Figures and Tables

**Figure 1 tropicalmed-07-00227-f001:**
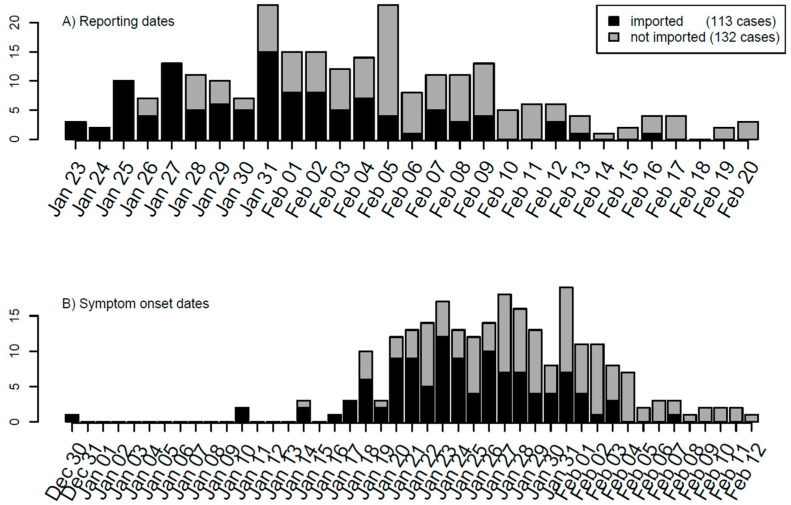
Timeline of the COVID-19 outbreak in Shaanxi province, China based on (**A**) reported dates, (**B**) dates of symptom onset, and (**C**) modified dates of symptom onset. For modelling’s sake, the daily numbers of imported and local cases have been marked separately.

**Figure 2 tropicalmed-07-00227-f002:**
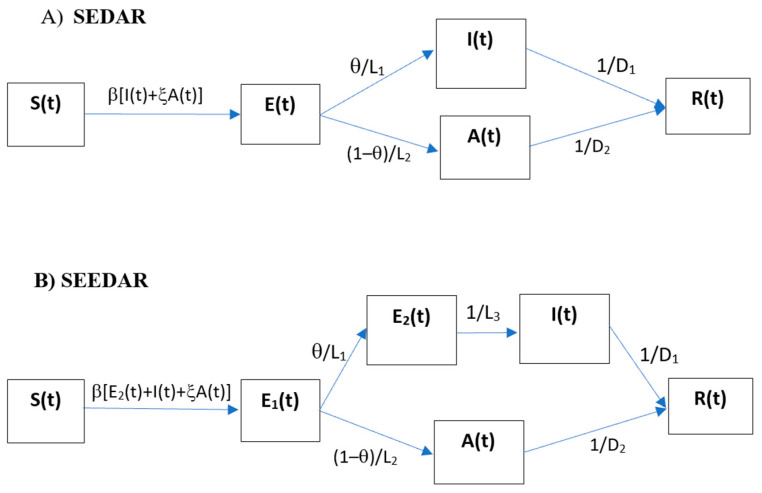
Flow chart of (**A**) SEDAR transmission model, (**B**) SEEDAR transmission model, and (**C**) SEEDDAAR transmission model.

**Figure 3 tropicalmed-07-00227-f003:**
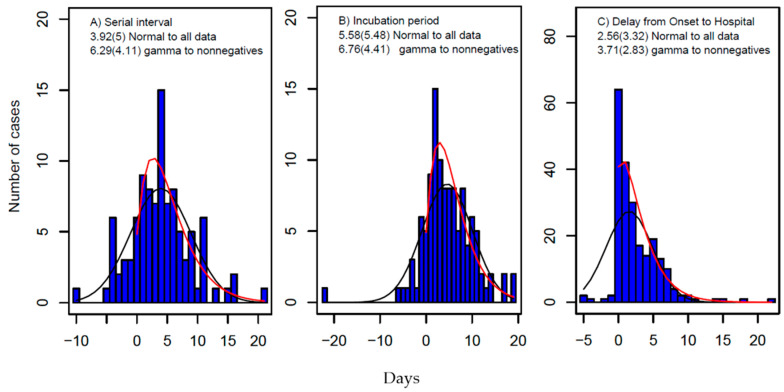
Distributions of (**A**) serial interval (SI), (**B**) incubation period, and (**C**) delay from date of symptom onset to hospital visit. The blue pillars represent the data, and the estimate of the mean and its standard deviation in brackets are obtained by fitting gamma distribution to nonnegative data (red curve) and normal distribution to all data (black curves shown in the graphs).

**Figure 4 tropicalmed-07-00227-f004:**
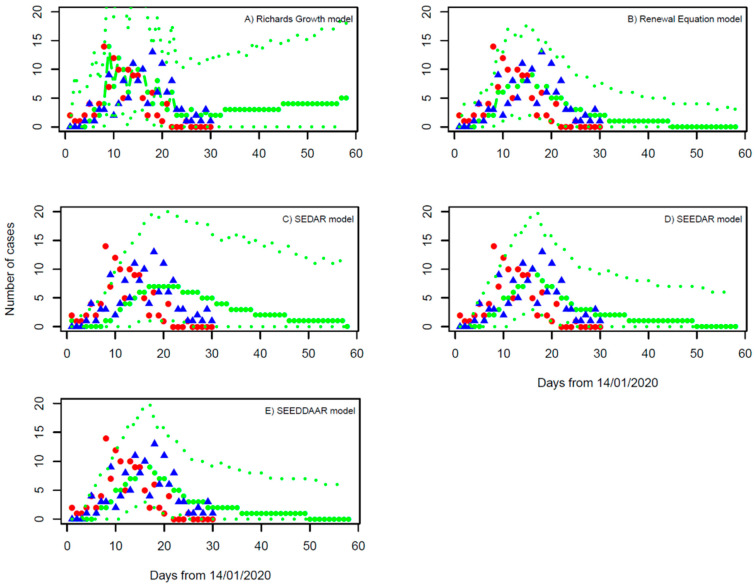
Fitting of (**A**) the Richards growth model, (**B**) the renewal equation model, (**C**) the SEDAR model, (**D**) the SEEDAR model, and (**E**) the SEEDDAAR model to outbreak data of symptom onset dates. Red dots represent the imported cases, and the blue triangles are the cases locally transmitted in Shaanxi province. The thick green line represents the median of MCMC samples, and the thin lines represent their upper and lower levels of 95% confidence intervals.

**Figure 5 tropicalmed-07-00227-f005:**
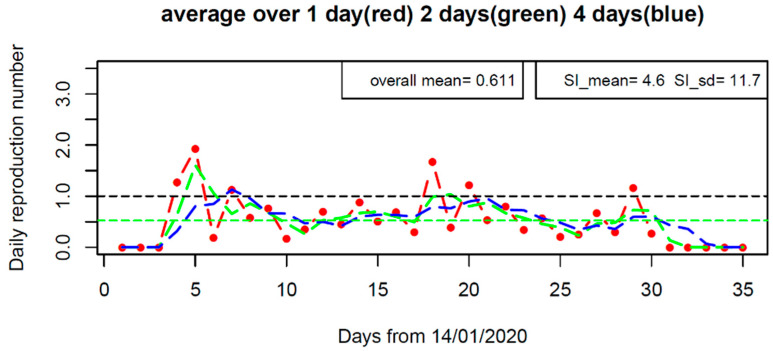
The effective reproductive number (*R*_t_) along the course of the outbreak in Shaanxi province, China under the SI distributions of SI_mean = 4.6 days, SI_sd = 11.7 days (the Maximum likelihood estimate of SI from model calibration of the renewal equation). *R*_t_ is evaluated by averaging over one, two, and four days.

**Table 1 tropicalmed-07-00227-t001:** Parameter estimates of five transmission models.

Parameter	Richards Growth		Renewal Equation		SEDAR		SEEDAR		SEEDDAAR
	Prior	Posterior	Prior	Posterior	Prior	Posterior	Prior	Posterior	Posterior
Growth rate (*r*)	[0,1.0]	0.02 [0.012,0.032]	–	–	–	–	–	–	–
Final epidemic size (*K*)	[1,6600]	3315 [56,6521]	–	–	–	–	–	–	–
Scaling exponent (*ν*)	[0.1,50]	24.51 [0.72,48.81]	–	–	–	–	–	–	–
Mean of SI (SI_mean)	–	–	U [3.5,10.0]	4.66 [3.53,7.18]	–	–	–	–	–
Standard deviation of SI (SI_sd)	–	–	U [3.0,15.0]	11.73 [5.85,14.88]	–	–	–	–	–
Transmission coefficient (β)	–	–	–	–	U [.001,0.5]	0.155 [0.117,0.186]	U [.001,0.5]	0.066 [0.029,0.154]	0.072 [0.032,0.180]
Latent period (*L*_1_) *	–	–	–	–	U [1.6,14.0]	1.81 [1.61,2.82]	U [1.0,10.0]	5.04 [1.25,9.65]	5.25 [1.28,9.76]
Pre-symptomatic infectious period (*L*_3_)	–	–	–	–	–	–	U [1.0,10.0]	1.45 [1.04,4.43]	1.45 [1.04,4.43]
Infectious period (*D*_1_) of diseased infections *	–	–	–	–	U [3.5,25.0]	3.75 [3.51,5.16]	U [1.5,15.0]	4.78 [1.61,14.06]	5.40 [1.68,13.97]
Dispersion parameter (*η*)	–	–	U [1.01,50.0]	1.58 [1.06,2.86]	U [1.01,50.0]	2.47 [1.56,4.431]	U [1.01,50.0]	1.73 [1.08,3.26]	1.71 [1.09,3.18]
*R* _0_ ^♦^	–	1.13 [1.08,1.21]	U [0.05,3.0]	0.61 [0.54,0.68]	–	0.59 [0.50,0.70]	–	0.45 [0.30,0.76]	0.53 [0.35,0.85]
DIC ^♣^	–	140.2	–	127.9	–	175.1	–	160.5	160.8

*: For three compartmental transmission models, the relative infectivity (*ξ*) of asymptomatic infections to symptomatic infection is set at 0.5, and the incubation and infectious period for asymptomatic infections are set to be equal to the counterparts of symptomatic infections (i.e., *L*_2_ = *L*_1_ and *D*_2_ = *D*_1_). As the proportion of asymptomatic infections is very small (i.e., 1 − *θ* = 1.1%), the other choices of these three parameters (say *ξ* = 1, *L*_2_ = 2*L*_1_ and *D*_2_ = 2*D*_1_) do not noticeably change the estimates of the model parameters listed here. The priors for the SEEDAR and SEEDDAAR models are the same. ^♦^: *R*_0_ for the Richards growth model is calculated via equation (2) with the gamma-distributed serial interval of mean = 6.29 days and SD = 4.11 days (shape parameter = 2.343, rate parameter = 0.372). ^♣^: Deviance information criterion (DIC) is a measure of model fitting.

## Data Availability

The data supporting reported results are provided in Figure 1 and Figure 3.

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
