# Peer review of "Importation, Local Transmission, and Model Selection in Estimating the Transmissibility of COVID-19: The Outbreak in Shaanxi Province of China as a Case Study"

_tropicalmed, 2022, doi:10.3390/tropicalmed7090227_

Round 1

Reviewer 1 Report

The manuscript is very interesting and provides valuable information to continue understanding the transmission process of COVID-19. However, I indicate some points to improve.
1) The frequency distributions shown in Fig. 1 would be very relevant if they indicate the basic descriptive parameters such as coefficient of variation, modal and average value. Furthermore, it is clear that the bias is shown in the frequencies, so it would be very useful to perform a K-S goodness-of-fit test to describe the bias.
2) In figure 3, I suggest only leaving the legend of the y axis, in the first graph because it is redundant in the two subsequent ones, the same in the x-axis. Also, don't use the number symbol, it is more correct to use Number of cases, days should be Days.
3) In figure 4, I suggest the same regarding the x and y axes, avoid redundancy if the axes indicate the same information. In addition, if the title of the y-axis is too long, they can leave Daily number of cases and in the foot of the figure indicate what the axis refers to.
4) Figure 5, adjust axis titles, change daily to Daily, days to Days.

In general, the analysis with the models allows, albeit complex, the explanation of the estimation of the transmissibility of COVID-19.

Author Response

Many thanks to your positive comments. Here are our responses to your suggestions:

1) The frequency distributions shown in Fig. 1 would be very relevant if they indicate the basic descriptive parameters such as coefficient of variation, modal and average value. Furthermore, it is clear that the bias is shown in the frequencies, so it would be very useful to perform a K-S goodness-of-fit test to describe the bias.

1) Figure 1 is not a frequency distribution, it is the timeline of cases based on different time points: A) reported dates, B) dates of symptom onset and C) modified dates of symptom onset. The analyses suggested are not relevant.

2) In figure 3, I suggest only leaving the legend of the y axis, in the first graph because it is redundant in the two subsequent ones, the same in the x-axis. Also, don't use the number symbol, it is more correct to use Number of cases, days should be Days.

2) Done as suggested.

3) In figure 4, I suggest the same regarding the x and y axes, avoid redundancy if the axes indicate the same information. In addition, if the title of the y-axis is too long, they can leave Daily number of cases and in the foot of the figure indicate what the axis refers to.

3) Done as suggested.

4) Figure 5, adjust axis titles, change daily to Daily, days to Days.

4) Done as suggested.

Reviewer 2 Report

­The proposed manuscript is devoted to the results of a study related to estimating the transmissibility of COVID-19 considering the outbreak in Shaanxi Province of China during the first months of the pandemic.

Preliminaries to the research problem are provided. In particular, the authors mention the importance of improving the understanding of the transmissibility of COVID-19 and the impact of control measures implemented on stopping the spread. The authors review different models proposed for analysis of the transmission dynamics of COVID-19. They bring out the necessity  of finding such a model that can provide a simple and accurate tool for estimation of the essential epidemiological parameters allowing thus to predict the trend of spread within a population.

The authors propose four transmission models describing the statistical data from the early Shaanxi outbreak in China. An important characteristic of the proposed models is that they distinguish imported and local cases in their contribution to local transmission dynamics.

Details related to the data as well as used variables are provided. The authors describe carefully the proposed models and inference methods. They demonstrate how to select a reasonable model and correctly use the outbreak data. The authors use Markov chain Monte Carlo sampling for parameter estimation.

The results of the analysis of the proposed models are presented. They are compared between themselves as well as with results of other models, which mix up imported cases with local cases. It is shown that the proposed models give better results and successfully predict the actual epidemic situation in Shaanxi province from the late of February 2020.

The presentation of the main results is clear and comprehensive. From a formal point of view, all the contents seem to be correct. The results are valuable and worthy of being published taking into account their possible applications in medicine and epidemiology for analysis, prediction and control of the spread of various infections.

Minor revisions are suggested to improve the quality of the exposition:

p. 1 line 38: I suggest to write ”(denoted as R_0) or ”(denoted here as R_0)” instead of ”(denoted it as R_0)”.

p. 2 line 98: I suggest to write ”with the later one being infectious” instead of ”with the late one being infectious”.

p. 4 lines 153, 159, 164 etc.: I suggest to use “,” or “.” at the end of each formula. Please double check all formulas.

Author Response

Many thanks to your positive comments.

1. line 38: I suggest to write ”(denoted as R_0)” or ”(denoted here as R_0)” instead of ”(denoted it as R_0)”

Done as suggested

2. line 98: I suggest to write ”with the later one being infectious” instead of ”with the late one being infectious”.

Replaced with " with the latter one being infectious,"

3. lines 153, 159, 164 etc.: I suggest to use “,” or “.” at the end of each formula. Please double check all formulas.

Done as suggested

All your suggestions have been taken, except that ”with the late one being infectious” is replaced by ”with the latter one being infectious”.

Reviewer 3 Report

I suggest to add one more compartmental model in which not only several exposed classes are present (such as in the SEEDAR model) but there are also  muliple infectious classes assume a gamma-distributed infectious period. Such models have been used in several papers (see a couple of examples in the references below) and have been shown to follow more precisely the empirical observations. 

[1] Pasetto D, Lemaitre JC, Bertuzzo E, Gatto M, Rinaldo A. Range of reproduction number estimates for COVID-19 spread. Biochem Biophys Res Commun. 538(2021), 253-258.

[2] Xiang Y, Jia Y, Chen L, Guo L, Shu B, Long E,COVID-19 epidemic prediction and the impact of public health interventions: A review of COVID-19 epidemic models, Infectious Disease Modelling 6(2021), 324-342.

[3] G. Röst, F. A. Bartha, N. Bogya, P. Boldog, A. Dénes, T. Ferenci, K. J. Horváth, A. Juhász, Cs. Nagy, T. Tekeli, Zs. Vizi, B. Oroszi, Early phase of the COVID-19 outbreak in Hungary and post-lockdown scenarios, Viruses, 12(2020) No. 7, 708.

[4] M. Polver, F. Previdi, M. Mazzoleni, A. Zucchi,  A SIAT3 HE model of the COVID-19 pandemic in Bergamo, Italy, IFAC PapersOnLine 54-15 (2021) 263–268.

[5] D. Champredon, J. Dushoff, D. J. D. Earn, Equivalence of the Erlang-distributed SEIR epidemicmodel and the renewal equation, SIAM J. Appl. Math., 78(2018), 3258-3278.

Author Response

I suggest to add one more compartmental model in which not only several exposed classes are present (such as in the SEEDAR model) but there are also  muliple infectious classes assume a gamma-distributed infectious period. Such models have been used in several papers (see a couple of examples in the references below) and have been shown to follow more precisely the empirical observations. 

Many thanks for the good suggestion. To allow the infectious periods to follow gamma distribution, the corresponding compartments (i.e., I and A) have been divided evenly into two parts (i.e. I1 and I2, A1 and A2), so we have the fifth model: SEEDDAAR compartmental model. The relevant contents have been added in the following places: Lines 12,18,96-98,152-153 Lines 290-309 2.2.5 SEEDDAAR transmission model Lines 316,323 Lines 464-470 3.2.5 SEEDDAAR model Lines 483,488-493. Figure 2– Flow chart of SEEDDAAR model are added Table 1– Results of SEEDDAAR model are added Lines 292-294: six references are added: Wearing et al 2005; Champredon et al 2018; Polver et al 2021; Röst et al 2020; Pasetto et al 2021; Xiang et al 2021 Champredon et al is added on line 501 Xiang et al is added on line 545. The number of references has been rearranged.